# Validity of a 6-item movement control test battery for evaluation of movement control impairment in the lumbar spine

**Peemongkon Wattananon** [1]*, **Aminu Alhassan Ibrahim**[2], **Sasithorn Kongoun**[1], **Katayan Klahan**[1]

**1** Spine Biomechanics Laboratory, Faculty of Physical Therapy, Mahidol University, Nakhon Pathom, Thailand, **2** Physiotherapy Department, Tishk International University, Erbil, Kurdistan Region, Iraq

* peemongkon.wat@mahidol.ac.th

## Abstract

### Background

Patients with chronic non-specific low back pain (CNLBP) demonstrate movement control impairment (MCI) based on a 6-item motor control test (MCT) battery, suggesting its potential clinical utility.

### Objectives

This study aimed to determine the criterion-related validity of the 6-item MCT battery in discriminating MCI among individuals with CNLBP, a history of low back pain (HxLBP), and no low back pain (NoLBP).

### Methods

One hundred forty-one participants aged 20–40 years (47 participants per group) were recruited. The MCT battery (the waiter's bow, pelvic tilt, sitting knee extension, quadruped rocking forward, quadruped rocking backward, prone knee flexion) was rated using different rating methods, including individual tests, summation, and direction-specific tests. A 2x2 contingency table using a known group against the rating result was used to calculate chi-square, sensitivity, specificity, and positive and negative likelihood ratios for each pair separately.

### Results

Chi-square tests demonstrated significant associations ($P < 0.05$) between MCT and CNLBP when compared against NoLBP. In addition, the waiter's bow and flexion-specific tests demonstrated high sensitivity (81% and 72%, respectively), while sitting knee extension, prone knee flexion, and summation demonstrated high specificity (70%, 70%, and 89%, respectively).

**Data availability statement:** All relevant data are within the article and its Supporting Information files.

**Funding:** This study was funded by the National Research Council of Thailand (Grant No. N42A650360) awarded to PW. The authors also acknowledge support from the International Postdoctoral Fellowship 2024, Mahidol University. The funders had no role in study design, data collection and analysis, decision to publish, or preparation of the manuscript.

**Competing interests:** The authors have declared that no competing interests exist.

**Abbreviation:** CI, confidence interval; CNLBP, chronic non-specific low back pain; HxLBP, history of low back pain; MCI, movement control impairment; MCT, movement control test; NLR, negative likelihood ratio; NoLBP, no low back pain; NPRS, numeric pain rating scale; ODI, Oswestry disability index; PABAK, prevalence-adjusted and bias-adjusted kappa; PLR, positive likelihood ratio; SN, sensitivity; SP, specificity.

## Conclusion

Findings suggest the acceptable validity of the MCT battery when compared between CNLBP and NoLBP. The waiter's bow and flexion-specific tests effectively ruled out individuals with NoLBP with negative results, while sitting knee extension, prone knee flexion, and summation effectively ruled in the individuals with CNLBP with positive results. These findings highlight the clinical utility of these tests in assessing MCI in the lumbar spine.

## Introduction

Low back pain (LBP) remains the most prevalent musculoskeletal disorder and the main cause of global disability [1,2]. It causes significant personal and societal burdens due to its multifactorial impact, including activity limitation, work absenteeism, lost productivity, and reduced quality of life [3,4]. As the current projection indicates that more than 800 million people will have LBP by 2025 [2], the impact of LBP may be even more dire, especially if effective treatment methods are not identified and implemented.

Non-specific LBP represents 90%–95% of cases [5], with chronic non-specific LBP (CNLBP) accounting for the majority of disability-related LBP [6]. The efficacy of current treatment techniques varies considerably, perhaps because individuals with CNLBP are highly heterogeneous [7], underscoring the importance of subgrouping [8–10]. One subgroup typically seen among this population is the movement control impairment (MCI) [9,11]. This subgroup presents with clinically observed aberrant movement patterns during functional movements, which are thought to contribute to the onset of injury and recurrent episodes of LBP [12].

The diagnosis of CNLBP does not typically require diagnostic imaging unless serious underlying conditions are suspected [13,14]. Clinical tests and assessments play a crucial role in identifying specific impairments [9,15]. However, the validity of these tests remains a challenge, partly due to the absence of a universally accepted gold standard for identifying MCI [15,16]. This limitation has been a significant barrier in validity-related research,

Several clinical screening or movement control tests (MCTs) for MCI have been proposed and evaluated [10,12,17–26]. Notable MCTs include those evaluated by Luomajoki et al. [24], who assessed the reliability of a standardized test battery consisting of 10 active movement tests for the lumbar spine, with 6 items (waiter's bow, pelvic tilt, sitting knee extension, quadruped rocking forwards, quadruped rocking backward, and prone knee flexion) demonstrating adequate reliability [24]. Additionally, acceptable reliability consisting of the 6-item MCT was reported as summation and direction-specific grading methods [26,27], highlighting their potential clinical utility.

While the reliability of the 6-item MCT battery and its grading methods has been supported with varying degrees of acceptable agreement [24,26,27], its validity has not been thoroughly established. A previous study [18] used the summation score

of the 6-item MCT battery and reported that patients with CNLBP had an average of 2.21 positive MCTs (95% CI: 1.94–2.48) compared to 0.75 (95% CI: 0.55–0.95) for asymptomatic individuals, suggesting its potential ability to differentiate between groups. However, further validation is required to confirm its discriminatory ability in distinguishing individuals with CNLBP, a history of LBP (HxLBP), and no LBP (NoLBP).

Thus, the primary objective of this study was to determine the criterion-related validity of the 6-item MCT battery using different grading methods (individual tests, direction-specific tests, and test summation) in discriminating MCI in the lumbar spine among individuals with CNLBP, HxLBP, and NoLBP.

## Methods

### Study design

This cross-sectional study used a convenience sample of participants with CNLBP, HxLBP, and NoLBP and was conducted at the Faculty of Physical Therapy, Mahidol University. This human research followed the principles of the Declaration of Helsinki. The study was approved by the Mahidol University Institutional Review Board (COA No. MU-CIRB 2020/084.1806) in June 2020. The study purpose and procedure were explained to eligible participants, who were then given informed consent forms to sign, seeking their full participation and willingness to partake. Data were collected between January 1 and September 30, 2023.

### Participants

One hundred forty-one male and female participants aged 20–40 were recruited into the study, forming three known groups with equal sample sizes (47 for each group) for the validity evaluation. The equal group sizes (47 participants per group) were achieved by recruitment using a convenience sample. The research manager recruited participants until the target sample size (47 per group) was reached, ensuring balanced group sizes.

Although inter-rater reliability was not a primary objective of this study, ensuring that the results were not influenced by inadequate reliability was essential. To address this, the research manager used a computer-generated random selection process to identify a subset of 47 participants from the total sample (141 participants). If a recruited participant's assigned number matched a number in the computer-generated list, the research manager arranged for two raters to simultaneously observe and score the participant's performance on the 6-item MCT battery. Given the time and cost constraints, it was not feasible to determine inter-rater reliability across 141 participants. Instead, this subset was used to establish the inter-rater reliability while maintaining efficiency in data collection.

Participants with CNLBP were included at the time of the study if they had an active episode of LBP or recurrent episodes for ≥ 3 months, while participants with HxLBP were included if they had a recurrent pattern of LBP with at least two episodes that interfered with activities of daily living and/or required treatment, but currently pain-free [28,29]. Participants with NoLBP were included if they had no previous history of LBP for more than six months. The exclusion criteria included participants with back-related infections, tumors, or cancer; serious spinal pathologies such as disc herniation, spondylolisthesis, spinal fracture, or spinal stenosis; serious neurological symptoms; history of spinal surgery; acute LBP; pain in other areas of the body that would alter participants' movement patterns; restricted active range of motion due to joint stiffness preventing completion of the test; and pregnancy.

This study utilized data from an ongoing research project with a pre-determined sample size of 141 participants (47 participants per group). Since no previous studies have investigated the validity of the 6-item MCT battery, a direct sample size calculation based on previous findings was not possible. However, to ensure that the available sample size was adequate for detecting meaningful effects, we performed a post-hoc sample size analysis using G*Power (version 3.1.9.7 for Windows). A sample size calculation was conducted based on a medium effect size ($\rho = 0.3$), a confidence level of 0.05, and 80% power using a chi-square test. The results indicated that a minimum of 41 participants per group would be required to detect a medium effect size. Given that our study included 47 participants per group, the sample

size exceeded this minimum requirement, confirming that the study was sufficiently powered to detect the intended effect.

## Demographic and clinical characteristics

A Google form questionnaire was used to collect de-identified demographic data (e.g., age, gender, height, weight, and body mass index) and clinical characteristics (e.g., pain during the episode, duration since the first episode of LBP, frequency of recurrent pain during six months, and disability level). The participants were asked to complete the form themselves.

Pain intensity during the episode of LBP was evaluated using the numeric pain rating scale (NPRS), which consists of scores ranging from 0 (no pain) to 10 (worst pain) [30]. The NPRS has been shown to be a valid and reliable measure of pain intensity in patients with LBP [8]. Disability due to LBP was evaluated using the Oswestry disability index (ODI). It is a valid and reliable tool to measure disability in patients with LBP [8]. It consists of 10 items with a score level of 0–5 scale [30]. The total possible score was then multiplied by 100 and presented as a percentage, with higher scores indicating severe disability [31]. The ODI scores were categorized as minimal disability (0–20), moderate disability (21–40), severe disability (41–60), crippled (61–80), and bed-bound (81–100) [30].

## Raters

Before data collection, two raters with 10 and 3 years of experience underwent the training program that involved 24 hours of supervised training under a physical therapist with over 20 years of experience in the musculoskeletal system. It was divided into three sessions (8 hours for each session). The expert provided a lecture on test descriptions and grading criteria with examples in the first session. In the second session, two raters observed participants performing the MCT battery under the expert's guidance. Any discrepancies were discussed and resolved. In the final session, the raters independently performed clinical observations. The agreement between the expert and raters was over 70%, meeting the criteria for proceeding to actual data collection [27]. A physical therapist with 10 years of clinical experience in musculo-skeletal physical therapy performed the 6-item MCT battery on 141 participants for validity evaluation, while 47 out of 141 participants were randomly selected and rated by a physical therapist with 3 years of experience to establish inter-rater reliability. The raters were blinded to the participants' demographic and clinical characteristics.

## Performance of the 6-item movement control test battery

The MCT battery includes 6 active movement tests (Fig 1) evaluating MCI of the lumbar spine, which consists of 3 flexion-specific tests (waiter's bow, sitting knee extension, and quadruped rocking backward) and 3 extension-specific tests (pelvic tilt, prone knee flexion, and quadruped rocking forward) [27]. The participants were instructed to perform three repetitions for each test. To evaluate the individual tests, a test was considered "negative" if participants performed it correctly for all three repetitions. Conversely, a test was deemed "positive" if participants could not perform it correctly for two or more repetitions. For tests involving both legs, such as the sitting knee extension and prone knee flexion tests, a positive result on either leg was considered a "positive" overall result for that test. These individual test results were then used for both the summation and direction-specific grading methods. In the summation method, if two or more individual tests yielded a positive result, the overall rating was considered "positive". Otherwise, it was rated as "negative" [27]. The direction-specific grading method was rated as "positive" when at least two out of three direction-specific tests were positive. Otherwise, it was rated as "negative" [27].

## Statistical analyses

Statistical analyses were performed using SPSS version 23.0 (SPSS Inc., Chicago, IL, USA) and Microsoft Excel (Microsoft 365, Microsoft Corp., Redmond, WA, USA). A one-way ANOVA with post-hoc Bonferroni was used to determine

## Flexion-specific tests

Negative Positive

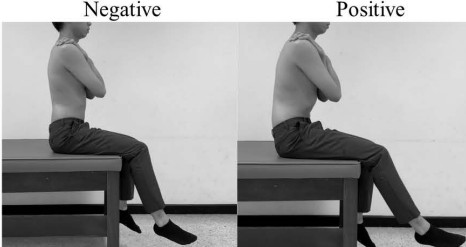

**Waiter's bow**: Hip flexion in standing upright position (both knees in slightly flexion) without lower back movement (flexion). The test is considered negative if the participant can perform a 30-degree forward bending of the hip without lower back movement. If the lumbar flexion occurs, the test is positive.

Negative Positive

**Sitting knee extension**: In a sitting upright position with lumbar normal lordosis, straighten one knee without lower back movement. The test is considered negative if the participant can perform knee extension without lower back movement, while it is positive if lumbar flexion occurs.

Negative Positive

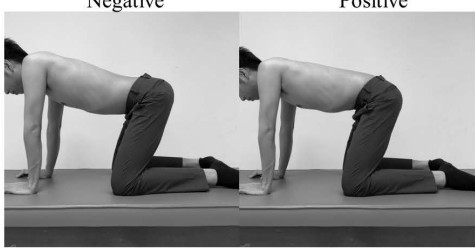

**Quadruped rocking backward**: Rocking backward in a quadruped position for 30 degrees of hip flexion from the starting position. The test is negative if the participant can perform without lower back movement, while it is positive when hip flexion causes lumbar flexion.

## Extension-specific tests

Negative Positive

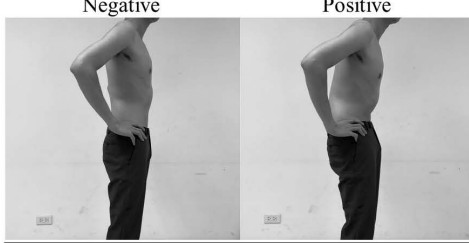

**Pelvic tilt**: Posterior pelvis tilt actively in a standing upright position (hand-on pelvis). The test is negative when the lumbar spine is neutral during active posterior pelvic tilt, while the thoracic spine or knee flexion is considered positive.

Negative Positive

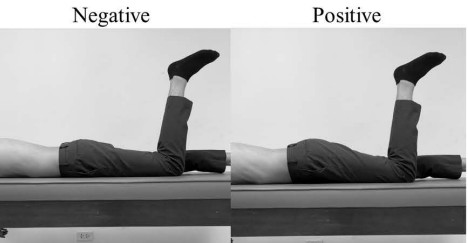

**Prone knee flexion**: Active single knee flexion in a prone lying position. The test is negative if the participant can perform active knee flexion to 90 degrees without lower back movement. If the lumbar extension or rotation movement occurs, the test is positive.

Negative Positive

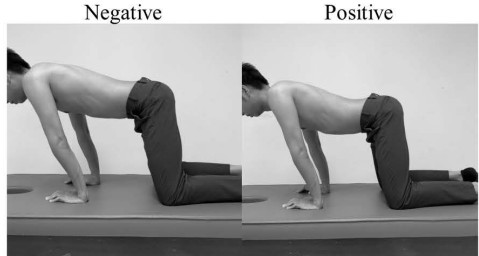

**Quadruped rocking forward**: Rocking forward in a quadruped position to 30 degrees of hip extension from the starting position. The test is negative if the participant can perform without lower back movement, while it is positive when hip movement leads to the extension of the lower back.

**Fig 1. Movement control test battery.**

differences in age and BMI among the three groups, while an independent t-test determined differences in clinical data between CNLBP and HxLBP groups. A chi-square test ($\chi^2$) assessed sex proportion differences among the three groups.

To evaluate the inter-rater reliability, the $\chi^2$ test was used to determine the association between the two raters. The kappa coefficient ($\kappa$) and prevalence-adjusted and bias-adjusted kappa (PABAK) were computed along with their 95% confidence intervals (CI). Kappa and PABAK values were interpreted as follows: < 0 indicating poor, 0–0.20 slight,

0.21–0.40 fair, 0.41–0.60 moderate, 0.61–0.80 substantial, and 0.81–1.00 excellent to almost perfect agreement. Generally, a κ or PABAK value of >0.4 was considered adequate reliability. For the rating summation method, we used the intraclass correlation coefficient ($ICC_{2,k}$) and the standard error of the mean to represent the inter-rater reliability in addition to kappa statistics. The ICC values are <0.5, indicating poor, 0.5–0.75 moderate, 0.76–0.9 good, and >0.9 excellent reliability.

For criterion-related validity, each pair of groups was analyzed separately to understand their performance, given the unclear similarity between the HxLBP group and either the NoLBP or CNLBP groups. The χ² test was used to determine the association between test results and the known groups through a 2x2 contingency table, which was further analyzed in conjunction with sensitivity (SN), specificity (SP), positive and negative likelihood ratios (PLR, and NLR, respectively). SN and SP values were interpreted as high (≥90%), moderate (70–89%), and low (<70%). The PLR values were interpreted as small (2.0 to 5.0), moderate (5.0 to 10.0), and large (>10), while NLR values were interpreted as small (0.2 to 0.5), moderate (0.1 to 0.2), and large (<0.1) [32]. However, due to the absence of a gold standard for identifying MCI and its non-life-threatening nature, we considered a threshold of ≥60% to be acceptable for the SN and SP, and a threshold of >1.0 for PLR and <1.0 for NLR [33,34]. The significance level was set at 0.05 for all statistical analyses.

## Results

### Demographic and clinical characteristics

Demographic and clinical data are presented in Table 1. A total of 141 participants were included in the criterion-related validity analysis, with 47 participants per group: CNLBP (n=47), HxLBP (n=47), and NoLBP (n=47). The mean age for each group was 27.4±4.0 years (CNLBP), 29.6±5.8 years (HxLBP), and 29.1±5.7 years (NoLBP). Each group had more females than males, with sex distributions of 37 females and 10 males (CNLBP), 29 females and 18 males (HxLBP), and 34 females and 13 males (NoLBP). The CNLBP and HxLBP groups had moderate pain during the episode (4.8±1.9 and 2.7±1.4, respectively) and minimal disability scores (7.1±3.9% and 6.9±2.3%, respectively). There were no significant differences between the groups in demographic and clinical characteristics (P>0.05), except for the frequency of recurrent pain in the past 6 months (P=0.018).

**Table 1. Demographic and clinical characteristics of the participants for test-retest reliability (n=47) and diagnostic accuracy (n=141).**

| Variable | Inter-rater reliability | | | | Diagnostic accuracy | | | |
|---|---|---|---|---|---|---|---|---|
| | CNLBP (n=14) | HxLBP (n=18) | NoLBP (n=15) | *P*-value | CNLBP (n=47) | HxLBP (n=47) | NoLBP (n=47) | *P*-value |
| Age (years) | 30.2±6.0 | 29.9±6.0 | 30.2±5.4 | 0.957 | 27.4±4.0 | 29.6±5.8 | 29.1±5.7 | 0.515 |
| Sex (female/male) | 10/4 | 13/5 | 8/7 | 0.457 | 37/10 | 29/18 | 34/13 | 0.185 |
| Body mass index (kg/m²) | 22.6±4.0 | 22.0±4.12 | 23.4±4.6 | 0.667 | 24.0±4.9 | 23.1±4.3 | 23.0±4.0 | 0.458 |
| Pain during the episode (out of 10) | 4.6±1.5 | 4.0±1.7 | N/A | 0.087 | 4.8±1.9 | 4.7±1.4 | N/A | 0.838 |
| Duration since the first episode of low back pain (months) | 17.6±23.5 | 12.3±13.2 | N/A | 0.188 | 21.2±16.0 | 18.4±15.9 | N/A | 0.637 |
| Frequency of recurrent pain in the past 6 months (repetitions) | 41.2±52.5 | 15.6±37.1 | N/A | 0.011* | 84.9±72.1 | 28.0±56.6 | N/A | 0.018* |
| Oswestry disability index (percentage) | 8.1±6.2 | 7.2±6.7 | N/A | 0.514 | 7.1±3.9 | 6.9±2.3 | N/A | 0.867 |

CNLBP=chronic non-specific low back pain; HxLBP=history of low back pain; NoLBP=no low back pain; N/A=not applicable

*Significant difference between groups (*P*<0.05).

For inter-rater reliability, we randomly selected a subset of 47 participants from the total sample (CNLBP; n = 14, HxLBP; n = 18, NoLBP; n = 15). The mean age of the subset was comparable across groups (30.2 ± 6.0 years (CNLBP), 29.9 ± 6.0 years (HxLBP), and 30.2 ± 5.4 years (NoLBP), $P = 0.957$). The sex distribution for this subset was 10 females and 4 males (CNLBP), 13 females and 5 males (HxLBP), and 8 females and 7 males (NoLBP). There were no significant differences between the CNLBP and HxLBP groups in demographic and clinical characteristics ($P > 0.05$), except for the frequency of recurrent pain in the past 6 months ($P = 0.011$).

## Inter-rater reliability (n=47)

As shown in Table 2, the chi-square demonstrated a significant association ($P < 0.05$) between the two raters across rating methods. The results showed a fair to substantial inter-rater reliability (κ = 0.37–0.67, PABAK = 0.36–0.75) for the individual tests, fair inter-rater reliability (κ = 0.35, PABAK = 0.62) for the summation, and fair and substantial inter-rater reliability (κ = 0.69 and 0.35, PABAK = 0.70 and 0.32) for the flexion and extension (direction-specific grading). Most of the grading methods had acceptable κ of > 0.4 except for the prone knee flexion test (0.37), summation (0.35), and extension test (0.35). Table 3 presents mean scores for each group and rater, indicating more positive test results in the HxLBP and CNLBP groups. The inter-rater reliability for the summation method demonstrated good inter-rater reliability (ICC$_{2,k}$ = 0.803), while the SEM was 0.67.

## Criterion-related validity (n=141)

When comparing the CNLBP with the NoLBP group, chi-square tests demonstrated a significant association between positive test results and CNLBP ($P < 0.05$) for individual tests (except pelvic tilt and quadruped rocking backward tests), direction-specific tests, and test summation. The sensitivity values of the waiter's bow test (81%) and flexion-specific test (72%) were acceptable, while the sitting knee extension, prone knee flexion tests, and test summation showed acceptable specificity values (70%, 70%, and 89%, respectively). In addition, test summation demonstrated the highest PLR (3.40).

**Table 2. Inter-rater reliability of the 6-item movement control test battery using different rating methods.**

| Rating method | | χ² | P-value | κ (CI) | PABAK (CI) |
|---|---|---|---|---|---|
| Individual (Positive ≥ 2 out of 3 repetitions) | Waiter's bow | 21.05 | <0.001 | 0.67 (0.42–0.91) | 0.75 (0.56–0.94) |
| | Sitting knee extension | 15.50 | <0.001 | 0.57 (0.34–0.81) | 0.57 (0.35–0.81) |
| | Quadruped rocking backward | 13.44 | <0.001 | 0.53 (0.29–0.77) | 0.53 (0.26–0.75) |
| | Pelvic tilt | 19.09 | <0.001 | 0.64 (0.39–0.88) | 0.70 (0.46–0.88) |
| | Prone knee flexion | 7.67 | 0.006 | 0.37 (0.13–0.62) | 0.36 (0.07–0.60) |
| | Quadruped rocking forward | 10.57 | 0.001 | 0.44 (0.18–0.71) | 0.53 (0.26–0.75) |
| MCT summation (Positive ≥ 2/6 tests) | | 5.93 | 0.015 | 0.35 (0.02-0.69) | 0.62 (0.35–0.81) |
| Direction-specific test (Positive ≥ 2/3 tests) | Flexion test | 23.63 | <0.001 | 0.69 (0.49–0.90) | 0.70 (0.46–0.88) |
| | Extension test | 7.70 | 0.006 | 0.35 (0.13–0.58) | 0.32 (0.07–0.60) |

χ² = chi-square test; κ = Cohen's kappa coefficient; PABAK = prevalence-adjusted and bias-adjusted kappa

**Table 3. Mean and standard deviation of sum scores for each group and rater.**

| Rater | NoLBP (n = 15) | HxLBP (n = 18) | CNLBP (n = 14) | All (n = 47) | ICC$_{2,k}$ | SEM |
|---|---|---|---|---|---|---|
| Rater 1 | 1.73 ± 1.16 | 3.06 ± 1.43 | 3.21 ± 1.48 | 2.68 ± 1.49 | 0.803 | 0.67 |
| Rater 2 | 1.73 ± 1.33 | 3.28 ± 1.13 | 4.21 ± 1.05 | 3.06 ± 1.52 | | |

NoLBP = no low back pain, HxLBP = history of low back pain, CNLBP = chronic non-specific low back pain, ICC = intraclass correlation coefficient, SEM = standard error of the mean

Although both direction-specific tests demonstrated significant associations, only the acceptable sensitivity was observed in the flexion-specific test.

According to the comparison between HxLBP and NoLBP groups, only the waiter's bow test, quadruped rocking forward test, and test summation were significantly associated ($P < 0.05$) with HxLBP. According to these tests, the waiter's bow test showed an acceptable sensitivity value (81%), while the quadruped rocking forward test and test summation showed an acceptable specificity value (87% and 94%, respectively). Additionally, test summation demonstrated the highest PLR (5.67). No significant associations ($P > 0.05$) were observed between CNLBP and HxLBP groups. The criterion-related validity of the 6-item MCT battery across the three groups is shown in Table 4.

## Discussion

To our knowledge, this is the first study to evaluate the validity of the 6-item MCT battery for identifying MCI, focusing on individual tests, direction-specific and summation gradings by comparing participants with CNLBP, HxLBP, and NoLBP. According to the reliability, our results indicated that the 6-item MCT battery had acceptable inter-rater reliability, except for the prone knee flexion test, extension-specific test, and test summation. According to the validity, our findings apparently demonstrated the ability of the 6-item MCT battery to differentiate between CNLBP and NoLBP, while its ability to differentiate between other pairs was still questionable.

A previous systematic review showed that the intra-rater reliability of clinical screening tests for MCI ranged between moderate and substantial agreement, whereas the inter-rater reliability ranged between poor and very good agreement [16]. This underscores the necessity for further inter-rater reliability studies to ensure consistent assessments by different clinicians, which is crucial for coordinated care and accurate screening. Results of the present study indicated that the 6-item MCT battery is reproducible for most tests, as evidenced by the adequate inter-rater reliability coefficients. This aligns with previous research on the reliability of a similar 6-item MCT battery for assessing MCI in the lumbar spine [24].

Valid tests are indispensable for clinicians in patient settings to evaluate treatment methods. However, their use must be guided by evidence. Assessing the sensitivity and specificity, along with the likelihood ratios of a clinical test, is crucial to evaluate its validity before it can be considered an appropriate clinical tool [35].

When comparing the CNLBP and NoLBP groups, significant associations suggest that individuals with CNLBP have altered movement control to a certain degree. In addition, the results showed that the waiter's bow and flexion-specific tests were the most sensitive when comparing the CNLBP group with the NoLBP group, implying that these tests effectively exclude or rule out participants with NoLBP who have negative results.

The lower NLR values obtained for these two tests further support their sensitivity, subsequently leading to their clinical utility. In practical terms, a lower NLR (<1.0) suggests that a negative test is less likely to occur in individuals with a condition compared to those without the condition [34]. For example, considering the NLR score (0.39) for the waiter's bow, the probability of a negative test result or having no MCI in participants with NoLBP is 61% less likely compared to the probability of a negative test result in participants with CNLBP.

On the other hand, the sitting knee extension test, prone knee flexion test, and test summation demonstrated acceptable specificity and higher PLR, suggesting their ability to confirm or rule in participants with CNLBP who have positive

**Table 4. Criterion-related validity results for each pair (n = 141: CLBP = 47, HxLBP = 47, NoLBP = 47).**

| Group | Test | χ² | P-value | SN | SP | PLR | NLR | Accuracy |
|---|---|---|---|---|---|---|---|---|
| CNLBP vs NoLBP | Waiter's bow | 9.29 | 0.040* | **0.81 [0.66–0.91]** | 0.49 [0.34–0.64] | 1.58 [1.16–2.16] | 0.39 [0.20–0.75] | 0.65 [0.54–0.74] |
| | Pelvic tilt | 1.55 | 0.213 | 0.62 [0.46–0.75] | 0.51 [0.36–0.66] | 1.26 [0.87–1.82] | 0.75 [0.47–1.19] | 0.56 [0.46–0.67] |
| | Sitting knee extension | 6.27 | 0.012* | 0.55 [0.40–0.70] | **0.70 [0.55–0.83]** | 1.86 [1.12–3.09] | 0.64 [0.44–0.92] | 0.63 [0.52–0.73] |
| | Prone knee flexion | 6.27 | 0.012* | 0.55 [0.40–0.70] | **0.70 [0.55–0.83]** | 1.86 [1.12–3.09] | 0.64 [0.44–0.92] | 0.63 [0.52–0.73] |
| | Quadruped rocking backward | 0.18 | 0.674 | 0.62 [0.46–0.75] | 0.43 [0.28–0.58] | 1.07 [0.77–1.50] | 0.90 [0.55–1.47] | 0.52 [0.42–0.63] |
| | Quadruped rocking forward | 2.44 | 0.118 | 0.38 [0.25–0.54] | **0.77 [0.62–0.88]** | 1.64 [0.87–3.08] | 0.81 [0.61–1.06] | 0.57 [0.47–0.68] |
| | Flexion | 9.73 | 0.002* | **0.72 [0.57–0.84]** | 0.60 [0.44–0.74] | 1.79 [1.21–2.64] | 0.46 [0.28–0.78] | 0.66 [0.55–0.75] |
| | Extension | 4.30 | 0.038* | 0.55 [0.40–0.70] | 0.66 [0.51–0.79] | 1.63 [1.01–2.61] | 0.68 [0.46–0.99] | 0.61 [0.50–0.71] |
| | Summation | 8.55 | 0.003* | 0.36 [0.23–0.51] | **0.89 [0.77–0.96]** | 3.40 [1.37–8.46] | 0.71 [0.56–0.91] | 0.63 [0.52–0.73] |
| HxLBP vs NoLBP | Waiter's bow | 4.21 | 0.040* | **0.81 [0.67–0.91]** | 0.38 [0.25–0.54] | 1.31 [1.01–1.71] | 0.50 [0.25–1.00] | 0.60 [0.49–0.70] |
| | Pelvic tilt | 0.04 | 0.833 | 0.62 [0.46–0.75] | 0.40 [0.26–0.56] | 1.04 [0.75–1.43] | 0.95 [0.57–1.57] | 0.51 [0.41–0.62] |
| | Sitting knee extension | 2.73 | 0.098 | 0.55 [0.40–0.70] | 0.62 [0.46–0.75] | 1.44 [0.93–2.25] | 0.72 [0.49–1.07] | 0.59 [0.48–0.69] |
| | Prone knee flexion | 0.68 | 0.409 | 0.55 [0.40–0.70] | 0.53 [0.38–0.68] | 1.18 [0.70–1.76] | 0.84 [0.55–1.27] | 0.54 [0.44–0.65] |
| | Quadruped rocking backward | 1.08 | 0.298 | 0.62 [0.46–0.75] | 0.49 [0.34–0.64] | 1.21 [0.84–1.73] | 0.78 [0.49–1.25] | 0.55 [0.45–0.66] |
| | Quadruped rocking forward | 8.06 | 0.005* | 0.38 [0.25–0.54] | **0.87 [0.74–0.95]** | 3.00 [1.31–6.89] | 0.71 [0.55–0.91] | 0.63 [0.52–0.73] |
| | Flexion | 3.69 | 0.055 | **0.72 [0.57–0.84]** | 0.47 [0.32–0.62] | 1.36 [0.99–1.88] | 0.59 [0.34–1.03] | 0.60 [0.49–0.70] |
| | Extension | 1.53 | 0.216 | 0.55 [0.40–0.70] | 0.57 [0.42–0.72] | 1.30 [0.85–1.98] | 0.78 [0.52–1.16] | 0.56 [0.46–0.67] |
| | Summation | 12.45 | <0.001 | 0.36 [0.23–0.51] | **0.94 [0.82–0.99]** | 5.67 [1.78–18.06] | 0.68 [0.54–0.86] | 0.65 [0.54–0.74] |
| CNLBP Vs HxLBP | Waiter's bow* | 1.08 | 0.298 | 0.62 [0.46–0.75] | 0.49 [0.34–0.64] | 1.21 [0.84–1.73] | 0.780.49–1.25] | 0.55 [0.45–0.66] |
| | Pelvic tilt | 1.07 | 0.301 | 0.60 [0.44–0.74] | 0.51 [0.36–0.66] | 1.22 [0.84–1.77] | 0.79 [0.51–1.24] | 0.55 [0.45–0.66] |
| | Sitting knee extension | 0.76 | 0.384 | 0.38 [0.25–0.54] | **0.70 [0.55–0.83]** | 1.29 [0.73–2.27] | 0.88 [0.66–1.18] | 0.54 [0.44–0.65] |
| | Prone knee flexion | 2.88 | 0.090 | 0.47 [0.32–0.62] | **0.70 [0.55–0.83]** | 1.57 [0.92–2.68] | 0.76 [0.55–1.05] | 0.59 [0.48–0.69] |
| | Quadruped rocking backward | 0.39 | 0.535 | 0.51 [0.36–0.66] | 0.43 [0.28–0.58] | 0.89 [0.61–1.29] | 1.15 [0.74–1.79] | 0.47 [0.36–0.57] |
| | Quadruped rocking forward | 1.80 | 0.180 | 0.13 [0.05–0.26] | **0.77 [0.62–0.88]** | 0.55 [0.22–1.35] | 1.14 [0.94–1.38] | 0.45 [0.34–0.55] |
| | Flexion | 1.54 | 0.215 | 0.53 [0.38–0.68] | 0.60 [0.44–0.74] | 1.32 [0.85–2.04] | 0.79 [0.53–1.15] | 0.56 [0.46–0.67] |
| | Extension | 0.72 | 0.396 | 0.43 [0.28–0.58] | 0.66 [0.51–0.79] | 1.25 [0.74–2.10] | 0.87 [0.63–1.20] | 0.54 [0.44–0.65] |
| | Summation | 0.55 | 0.460 | 0.06 [0.01–0.18] | **0.89 [0.77–0.96]** | 0.60 [0.15–2.37] | 1.05 [0.93–1.19] | 0.48 [0.37–0.58] |

CNLBP = chronic non-specific low back pain; HxLBP = history of low back pain; NoLBP = no low back pain; SN = sensitivity; SP = specificity; PLR = positive likelihood ratio; NLR = negative likelihood ratio; χ² = chi-square test.

Note: *=significant association (*P*<0.05); **bold**=acceptable sensitivity or specificity (≥0.70).

tests. Typically, a higher PLR (>1.0) suggests that a positive test result is more likely in participants with the condition [34], which corresponds to high specificity, where the test accurately identifies most individuals who truly have the condition.

Test summation result suggests that individuals with CNLBP have altered movement control, but they were likely to have MCI in the flexion direction. Therefore, intervention should focus on key lumbar stabilizing muscles to provide adequate stability during these dynamic tasks.

Regarding the comparison between HxLBP and NoLBP groups, the waiter's bow again demonstrated its clinical utility in ruling out participants with NoLBP who have a negative result due to adequate sensitivity and lower NLR values. In another vein, higher specificity and PLR values exhibited by the quadruped rocking forward test and test summation suggest their clinical utility for ruling in participants with HxLBP who have a positive result. For the comparison of the CNLBP group with the HxLBP group, none of the tests exhibited significant associations between positive test results and CNLBP, even though sitting knee extension, prone knee flexion, quadruped rocking forward, and test summation reached acceptable specificity. This finding is not surprising, considering MCI is not anticipated to occur more frequently among

individuals with HxLBP compared to those with CNLBP. Therefore, clinicians should be cautious when screening MCI in individuals with HxLBP, as they may perform the tests similarly to those with either CNLBP or NoLBP. Moreover, individuals with HxLBP may have fluctuating symptoms and other causes contributing to pain other than MCI, making it difficult to detect or isolate MCI as the primary factor.

It is worth noting that the only test exhibiting both adequate sensitivity and specificity is the flexion-specific test. This is essential for clinicians when utilizing this test, as it can accurately identify individuals with and without MCI. This balance could minimize identifying errors, leading to accurate classification, management, and patient outcomes. With regard to the implication of criterion-related validity in the present study, the specificity and PLR would be more useful since they confirm the presence of the condition. Given adequate inter-rater reliability, along with specificity and PLR for the flexion, sitting knee extension, and quadruped rocking forward tests, these tests can be considered the most reliable and valid for identifying MCI in individuals with CNLBP or HxLBP. Moreover, the adequate inter-rater reliability, sensitivity, and NLR obtained for Waiter's bow and flexion tests suggest these tests are reliable and valid for excluding MCI in individuals with CNLBP or HxLBP. However, the fair inter-rater reliability obtained for the extension-specific test could have contributed to the inadequate validity findings between groups.

## Strengths and limitations

Being the first study to determine the criterion-related validity of the 6-item MCT battery with known group comparison could be considered as the key strength of the present study. However, the study has some limitations which should be considered when interpreting the findings. For example, judgments used in rating the 6-item MCTs were based on clinical observation, which is often challenging. This could partly explain the low reliability in some of the tests evaluated. For the criterion-related validity, only one rater was used to rate the tests, which could be a source of potential bias in the judgment. However, our rater has been validated by an expert with a percentage of agreement greater than 70%, which is considered acceptable [27]. While sensitivity, specificity, and likelihood ratios were used in this study to determine the validity, more robust statistical analyses, such as the receiver operating curve, could provide a more comprehensive view of the 6-item MCTs over a range of thresholds. Additionally, the absence of prior studies on the criterion-related validity of the 6-item MCT battery limits direct comparisons of the present study's results with others. Regardless of the aforementioned limitations, the present study provides valuable information regarding the clinical identification of MCI using commonly employed battery tests, along with their reliability and validity.

## Conclusion

In conclusion, most of the tests in the 6-item MCT battery demonstrate acceptable inter-rater reliability. The negative results in individuals with NoLBP for waiter's bow and flexion-specific tests can effectively rule them out from CNLBP, while sitting knee extension, prone knee flexion, and summation tests can effectively rule in the individuals with CNLBP with positive results. These findings underscore the clinical utility of these tests for assessing MCI in the lumbar spine.

## Supporting information

**S1 File. Minimal data set.**
(XLSX)

## Acknowledgments

We would like to thank the physical therapists for their assistance in data collection and extend our gratitude to all the study participants for their full cooperation throughout the study. We also want to thank the International Postdoctoral Fellowship 2024, Mahidol University.

## Author contributions

**Conceptualization:** Peemongkon Wattananon.

**Data curation:** Peemongkon Wattananon, Sasithorn Kongoun, Katayan Klahan.

**Formal analysis:** Peemongkon Wattananon.

**Funding acquisition:** Peemongkon Wattananon.

**Methodology:** Sasithorn Kongoun, Katayan Klahan.

**Writing – original draft:** Peemongkon Wattananon, Aminu Alhassan Ibrahim.

**Writing – review & editing:** Peemongkon Wattananon, Aminu Alhassan Ibrahim.

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
