## [Decision Letter · Decision Letter 0]

PONE-D-24-56878

Validity of a 6-item movement control test battery for evaluation of movement control impairment in the lumbar spine

PLOS ONE

Dear Dr. Wattananon,

Thank you for submitting your manuscript to PLOS ONE. After careful consideration, we have decided that your manuscript does not meet our criteria for publication and must therefore be rejected.

Specifically:

We regret to inform you that, based on the reviewers' critiques, your work has been deemed unsuitable for publication in PLOS ONE. The comment from the reviewer(s) is included at the end of this letter.

I am sorry that we cannot be more positive on this occasion, but hope that you appreciate the reasons for this decision.

Kind regards,

Sohel Ahmed, BPT, MPT, MDMR

Academic Editor

PLOS ONE

Reviewers' comments:

Reviewer's Responses to Questions

**Comments to the Author**

1. Is the manuscript technically sound, and do the data support the conclusions?

Reviewer #1: Yes

Reviewer #2: Partly

Reviewer #3: Partly

Reviewer #4: Partly

2. Has the statistical analysis been performed appropriately and rigorously?

Reviewer #1: Yes

Reviewer #2: Yes

Reviewer #3: Yes

Reviewer #4: No

3. Have the authors made all data underlying the findings in their manuscript fully available?

Reviewer #1: Yes

Reviewer #2: Yes

Reviewer #3: Yes

Reviewer #4: No

4. Is the manuscript presented in an intelligible fashion and written in standard English?

Reviewer #1: Yes

Reviewer #2: No

Reviewer #3: Yes

Reviewer #4: Yes

Reviewer #1: The study title is of clinical significance and the validation of 6 items movement control test battery has shown the clinical utility for ruling out back pain. However, more than 70% of the study participants were female. It will be good to write the justification in the discussion correlating with other studies.

Reviewer #2: Good paper with good objective. However there are some points needed to modify. First, what is your gold standard? You should present it completely in introduction section and try to relate your tool with that gold standard. Second, how do you justify your sample size? It is not enough in your paper. Third, in result and discussion part it is better to mention sample size and explain the text.

Reviewer #3: Keywords: Please remove non-MeSH terms and replace them with appropriate MeSH terms.

Abstract:

There appears to be a discrepancy in the reported sample size. The manuscript states that there are three groups (CNLBP, HxLBP, NoLBP), each consisting of 41 participants, which should result in a total sample size of 123 participants (3 × 41 = 123). However, the manuscript mentions that the total sample size is 141 participants.

Please clarify whether:

The total sample size is indeed 123, and the reported 141 participants is a typographical error.

Each group actually contains a different number of participants, which should be explicitly stated in the methods section.

Ensuring accurate reporting of participant numbers is crucial for the study's transparency and reproducibility.

Methods:

The Ethics Committee approval date should be included.

The manuscript would benefit from being structured using the TREND checklist. The authors are encouraged to provide supplementary materials indicating relevant page-line references within the control list.

In line 8, the statement "To establish inter-rater reliability, 47 participants were randomly selected among these three groups" is inconsistent with the study design. Since group assignment depends on the presence or absence of chronic low back pain, randomization is not possible in this context.

Were the equal group sizes achieved by chance or through selection and exclusion criteria? If participants were selected to ensure equal group sizes, how many were excluded, and for what reasons? This should be clarified. Additionally, a flowchart (CONSORT diagram) should be included.

The definitions of low back pain groups should be supported with references (lines 10–14).

How was kinesiophobia assessed? If a questionnaire was used, details such as minimum and maximum scores, validity-reliability measures, cutoff values, and references for its use in the original language should be provided. This approach should be consistently applied to all questionnaires used in the study.

The training process was described as 24 hours divided into three sessions, but how this time was distributed among the sessions and how raters' competencies were assessed should be clarified.

A 70% agreement threshold was set before data collection. However, the rationale behind this cutoff is unclear. Was it based on previous studies, or was it arbitrarily chosen? This should be justified with references.

The second rater assessed only 47 out of 141 participants. Was this number determined statistically, or was it randomly chosen? The selection method should be specified.

Simply reporting agreement percentages does not sufficiently establish reliability. Additional statistical analyses, such as intraclass correlation coefficients (ICCs) or Bland-Altman plots, could further strengthen the reliability assessment.

By addressing these points, the manuscript will significantly improve its methodological transparency and scientific rigor.

Reviewer #4: This study is an important step toward validating movement control tests for LBP, but stronger methodology is needed to establish clinical credibility. Please find below what could be done to improve the current work:

- The authors mentioned the lack of a gold standard which an essential element to determine the criterion-related validity. The author could have used EMG or motion analysis systems.

- The study did not include Receiver Operating Characteristic (ROC) curve analysis, which is standard in diagnostic accuracy research.

- Only one rater evaluated validity (while two were used for reliability) introducing a risk of subjective bias.

**Do you want your identity to be public for this peer review?** For information about this choice, including consent withdrawal, please see our Privacy Policy

Reviewer #1: **Yes: ** Bishnu Dutta Acharya

Reviewer #2: **Yes: ** Laleh Abadi marand

Reviewer #3: No

Reviewer #4: No

- - - - -

---

## [Author Response · Author response to Decision Letter 1]

3 Apr 2025

Response letter to the reviewer's comments for the manuscript:

ID: PONE-D-24-56878

Title: Validity of a 6-item movement control test battery for evaluation of movement control impairment in the lumbar spine.

Dear reviewers,

We would like to thank you for the effort and time you spent reviewing our manuscript entitled “Validity of a 6-item movement control test battery for evaluation of movement control impairment in the lumbar spine”.

You can find all the modifications/changes in the Marked-up file.

Please find below our point-by-point response to your constructive comments in tabular form

Reviewer’s Comments #1

1. The study title is of clinical significance and the validation of 6 items movement control test battery has shown the clinical utility for ruling out back pain. However, more than 70% of the study participants were female. It will be good to write the justification in the discussion correlating with other studies

Response: Thank you very much for the commendation. As we stated in the limitations, there are no previous studies on the criterion-related validity of the 6-item MCT battery to compare with our study. Regarding the predominance of females, several epidemiological studies have reported a higher overall prevalence of low back pain in females compared to males. Clinical practice guidelines also recognize sex as a risk factor for low back pain, with one study indicating that females have nearly three times the risk of developing back pain compared to males. Therefore, our sample aligns with findings in the existing literature.

References

1. Delitto A, George SZ, Van Dillen L, Whitman JM, Sowa G, Shekelle P, et al. Low back pain. J Orthop Sports Phys Ther 2012;42(4):A1-57.

2. Hoy D, Bain C, Williams G, March L, Brooks P, Blyth F, et al. A systematic review of the global prevalence of low back pain. Arthritis Rheum 2012;64(6):2028-37.

3. Huerta M, Salazar A, Moral-Munoz JA. Trends in chronic neck and low back pain prevalence in Spain (2006-2020): differences by sex, age, and social class. Eur Spine J 2025.

Reviewer’s Comments #2

1. Good paper with good objective. However, there are some points needed to modify.

Response: Thanks for your commendation.

2. First, what is your gold standard? You should present it completely in introduction section and try to relate your tool with that gold standard.

Response: Thank you for this germane comment. Unfortunately, to date, there is no gold standard for identifying MCI, which is a significant barrier in validity-related research. However, a 6-item movement control test (MCT) battery has been proposed as reliable screening tests and found that summation score of the test battery was associated with chronic low back pain. This highlights the potential ability of the MCT battery to discriminate individuals with CNLBP, a history of LBP (HxLBP), and no LBP (NoLBP). As such, we significantly revised the introduction, as suggested.

The revision 1) explicitly states that no universally accepted gold standard exists for identifying MCI, 2) explains why the MCI battery is used as a reference for assessing MCI including the findings from previous studies that support its clinical relevance, and 3) emphasizes that while reliability has been established, its validity remains uncertain, justifying the need for this study.

Page/line no. (see marked version): Page 4, line 17-19; Page 5, line 5-12.

References

1. Luomajoki H, Kool J, de Bruin ED, Airaksinen O. Reliability of movement control tests in the lumbar spine. BMC Musculoskelet Disord 2007;8:90.

2. Luomajoki H, Kool J, de Bruin ED, Airaksinen O. Movement control tests of the low back; evaluation of the difference between patients with low back pain and healthy controls. BMC Musculoskelet Disord 2008;9:170.

3. Second, how do you justify your sample size? It is not enough in your paper.

Response: Thank you for this observation. We want to point out that we did not calculate the sample size. We used a pre-determined sample from an ongoing study. We also want to emphasize that the calculation was performed to confirm adequacy rather than determine recruitment numbers. Therefore, we revised this section as shown in marked version.

We also addressed the rationale for selecting a subset of total sample size to establish the inter-rater reliability.

Page/line no. (see marked version): Page 7, line 6-15; Page 6, line 10-18.

4. Third, in the result and discussion part it is better to mention the sample size and explain the text.

Response: Thank you for pointing out this suggestion. We have now added a statement regarding the sample size in the Results and Discussion sections (as a limitation), as you suggested.

Page/line no. (see marked version): Page 10, line 21-23 to Page 11 line 1-3; Page 11, line 8-14.

Reviewer’s Comments #3

1. Keywords: Please remove non-MeSH terms and replace them with appropriate MeSH terms.

Response: Thank you for your comment. We have now revised the keywords as Low back pain, Movement disorders, Postural control, Validity and reliability based on your suggestion.

Page/line no. (see marked version): Page 2, line 23.

2. Abstract: There appears to be a discrepancy in the reported sample size. The manuscript states that there are three groups (CNLBP, HxLBP, NoLBP), each consisting of 41 participants, which should result in a total sample size of 123 participants (3 × 41 = 123). However, the manuscript mentions that the total sample size is 141 participants.

Please clarify whether:

- The total sample size is indeed 123, and the reported 141 participants is a typographical error.

- Each group actually contains a different number of participants, which should be explicitly stated in the methods section.

- Ensuring accurate reporting of participant numbers is crucial for the study's transparency and reproducibility.

Response: Thank you for these pertinent observations, and we apologize for the error. It was overlooked. We were supposed to write 47 (3 × 47 = 141) instead of 41. We have now corrected this section. We have now clearly indicated the sample size for each group in the method section as you suggested.

Page/line no. (see marked version): Page 2, line 8; Page 6, line 7-18; Page 7, line 6-15.

3. Methods: The Ethics Committee approval date should be included.

Response: Thank you for pointing this out. We have now included the date of ethical approval.

Page/line no. (see marked version): Page 6, line 1.

4. Methods: The manuscript would benefit from being structured using the TREND checklist.

Response: Thank you for this valuable comment. In structuring our manuscript, we followed the STROBE (Strengthening the Reporting of Observational Studies in Epidemiology) checklist to ensure transparent and comprehensive reporting. However, as the journal did not explicitly require authors to upload a research checklist during the initial submission, we did not include it at that time. For this resubmission, we have now included the STROBE checklist (S1 Appendix) as part of the submission package to enhance clarity and adherence to reporting guidelines.

Page/line no. (see marked version): S1 Appendix

5. Methods: line 8, the statement "To establish inter-rater reliability, 47 participants were randomly selected among these three groups" is inconsistent with the study design. Since group assignment depends on the presence or absence of chronic low back pain, randomization is not possible in this context.

Response: Thank you for your comments. The participants for the reliability assessment were randomly selected from total sample of 141 regardless of their group, as all of them were assessed for movement control impairment using the 6-item movement control test battery. The goal was to evaluate whether the rater reliably determined the presence or absence of movement control impairment. Of note, the movement control impairment may also be present in the NoLBP group. We revised this statement in the marked version.

Page/line no. (see marked version): Page 6, line 10-18.

6. Methods: Were the equal group sizes achieved by chance or through selection and exclusion criteria? If participants were selected to ensure equal group sizes, how many were excluded, and for what reasons? This should be clarified. Additionally, a flowchart (CONSORT diagram) should be included.

Response: Thank you for your insightful question. The equal group sizes (47 participants per group) were achieved by recruitment using a sample of convenience. The research manager recruited participants until the target sample size (47 per group) was reached, ensuring balanced group sizes.

To minimize bias, the rater assessing the participants was blinded to group allocation. Therefore, no participants were excluded to achieve equal group sizes.

For the inter-rater reliability analysis, the research manager used a computer-generated random selection process to identify a subset of 47 participants from the total sample (141 participants). If a recruited participant’s assigned number matched a number in the computer-generated list, the research manager arranged for two raters to simultaneously observe and score the participant’s performance on the 6-item MCT battery.

This approach ensured that group sizes remained equal while maintaining blinding and randomization where necessary.

The CONSORT diagram is appropriate for randomized controlled trials, thus not necessary for the design of our study, which is purely observational.

Page/line no. (see marked version): Page 6, line 7-18.

7. Methods: The definitions of low back pain groups should be supported with references (lines 10–14).

Response: Many thanks for this comment. We have now added references for the definition of low back pain groups as you suggested.

Page/line no. (see marked version): Page 6, line 22.

References

1. Sornkaew K, Thu KW, Silfies SP, Klomjai W, Wattananon P. Effects of combined anodal transcranial direct current stimulation and motor control exercise on cortical topography and muscle activation in individuals with chronic low back pain: A randomized controlled study. Physiother Res Int 2024;29(3):e2111.

2. Wattananon P, Thu KW, Maharjan S, Sornkaew K, Wang HK. Cortical excitability and multifidus activation responses to transcranial direct current stimulation in patients with chronic low back pain during remission. Sci Rep 2023;13(1):16242.

8. Methods: How was kinesiophobia assessed? If a questionnaire was used, details such as minimum and maximum scores, validity-reliability measures, cutoff values, and references for its use in the original language should be provided. This approach should be consistently applied to all questionnaires used in the study.

Response: Thank you for pointing this out, and we apologize for the error. We did not assess kinesiophobia. It was overlooked and has now been removed. For the disability assessment using the Oswestry Disability Index, a brief discussion about the scale has already been provided in the methods section. We have now added a statement on the reliability and validity of all the questionnaires evaluated, with appropriate references, as you suggested.

Page/line no. (see marked version): Page 7, line 22-23; Page 8, line 1-2.

Reference

1. Delitto A, George SZ, Van Dillen L, Whitman JM, Sowa G, Shekelle P, et al. Low back pain. J Orthop Sports Phys Ther 2012;42(4):A1-57.

9. The training process was described as 24 hours divided into three sessions, but how this time was distributed among the sessions and how raters' competencies were assessed should be clarified.

Response: The 24-hour supervised training was divided into three sessions (8 hours for each session). In the first session, the expert provided a lecture on test descriptions and grading criteria and examples. In the second session, two raters observed participants performing the MCT battery under the expert's guidance. Any discrepancies were discussed and resolved. In the final session, the raters independently performed clinical observations.

For raters’ competencies, the agreement between the expert and raters greater than 70% was used as a cut-off for the competency. After the raters reached this cut-off point, they proceeded to actual data collection.

Page/line no. (see marked version): Page 8, line 10-11.

10. A 70% agreement threshold was set before data collection. However, the rationale behind this cutoff is unclear. Was it based on previous studies, or was it arbitrarily chosen? This should be justified with references.

Response: Many thanks for this comment. This cut-off was based on a previous study suggesting that in rater training, an agreement level of 70% or higher is considered acceptable to ensure that raters are adequately trained and consistent before moving to the data collection. We have now added a reference to support our statement as you suggested.

Page/line no. (see marked version): Page 8, line 15.

Reference

1. Tsunoda Del Antonio T, José Jassi F, Chaves TC. Intrarater and interrater agreement of a 6-item movement control test battery and the resulting diagnosis in patients with nonspecific chronic low back pain. Physiother Theory Pract 2023;39(8):1716-26.

11. The second rater assessed only 47 out of 141 participants. Was this number determined statistically, or was it randomly chosen? The selection method should be specified.

Response: Thanks for this observation. We have addressed this in the marked version as you suggested.

Page/line no. (see marked version): Page 6, line 10-18.

12. Simply reporting agreement percentages does not sufficiently establish reliability. Additional statistical analyses, such as intraclass correlation coefficients (ICCs) or Bland-Altman plots, could further strengthen the reliability assessment.

Response: Once again, thank you for the comments and suggestions. We have now added ICCs as you suggested to strengthen the reliability assessment as you suggested.

Page/line no. (see marked version): Page 25, Table 3

13. By addressing these points, the manuscript will significantly improve its methodological transparency and scientific rigor.

Response: Once again, thank you very much for your comments.

Reviewer’s Comments #4

1. This study is an important step toward validating movement control tests for LBP, but stronger methodology is needed to establish clinical credibility.

Response: Thank you very much for your comments.

2. The authors mentioned the lack of a gold standard which an essential element to determine the criterion-related validity. The author could have used EMG or motion analysis systems. Thank you for your valuable suggestion. While EMG is a useful tool for analyzing muscle activity, it is not appropriate for identifying movement control impairment (MCI) itself. EMG provides insights into neuromuscular activation patterns underlying impaired movement but does not directly assess movement quality or control, which is the focus of this study.

Response: Regarding motion analysis systems (i.e., motion tracking systems and inertial measurement units), while they can detect altered movement patterns, there is currently insufficient evidence supporting their use as a gold standard for identifying MCI. Given the lack of a universally accepted reference standard, these systems remain research tools rather than clinically established diagnostic methods.

Additionally, this study was designed as clinical research, with an emphasis on developing findings that can be directly applied in clinical practice. Using specialized research-grade equipment such as motion analysis systems may limit clinical feasibility, as these technologies are not widely accessible in routine clinical settings.

For these reasons, we relied on clinically applicable movement control tests, ensuring that the results are both relevant and practical for real-world clinical decision-making.

We revised the introduction to justify the use of the 6-item movement control test battery and validated it using a known-group approach.

Page/line no. (see marked version): Page 4, line 17-19; Page 5, line 5-12

3. The study did not include Receiver Operating Characteristic (ROC) curve analysis, which is standard in diagnos

---

## [Decision Letter · Decision Letter 1]

Validity of a 6-item movement control test battery for evaluation of movement control impairment in the lumbar spine

PONE-D-24-56878R1

Dear Dr. Wattananon,

We’re pleased to inform you that your manuscript has been judged scientifically suitable for publication and will be formally accepted for publication once it meets all outstanding technical requirements.

Kind regards,

Seyed Hamed Mousavi

Academic Editor

PLOS ONE

Additional Editor Comments (optional):

Reviewers' comments:

Reviewer's Responses to Questions

**Comments to the Author**

Reviewer #1: All comments have been addressed

Reviewer #2: All comments have been addressed

2. Is the manuscript technically sound, and do the data support the conclusions?

Reviewer #1: Yes

Reviewer #2: Yes

3. Has the statistical analysis been performed appropriately and rigorously?

Reviewer #1: Yes

Reviewer #2: I Don't Know

4. Have the authors made all data underlying the findings in their manuscript fully available?

Reviewer #1: Yes

Reviewer #2: Yes

5. Is the manuscript presented in an intelligible fashion and written in standard English?

Reviewer #1: Yes

Reviewer #2: Yes

Reviewer #1: The proposed study is of high clinical significance and the author has addressed the previous comments of the reviewers.

Reviewer #2: Thank you al of the comments have been addressed. How ever I haven not been convinced about sample size.

**Do you want your identity to be public for this peer review?** For information about this choice, including consent withdrawal, please see our Privacy Policy

Reviewer #1: **Yes: ** Bishnu Dutta Acharya

Reviewer #2: No

---

## [Editor Report · Acceptance letter]

PONE-D-24-56878R1

PLOS ONE

Dear Dr. Wattananon,

I'm pleased to inform you that your manuscript has been deemed suitable for publication in PLOS ONE. Congratulations! Your manuscript is now being handed over to our production team.

Kind regards,

on behalf of

Dr. Seyed Hamed Mousavi

Academic Editor

PLOS ONE